# Surgical Treatment for Colorectal Cancer Partially Restores Gut Microbiome and Metabolome Traits

Hirotsugu Shiroma,[a] Satoshi Shiba,[b] Pande Putu Erawijantari,[a,c] Hiroyuki Takamaru,[d] Masayoshi Yamada,[d] Taku Sakamoto,[d] Yukihide Kanemitsu,[e] Sayaka Mizutani,[a] Tomoyoshi Soga,[f] Yutaka Saito,[d] Tatsuhiro Shibata,[b] Shinji Fukuda,[f,g,h] Shinichi Yachida,[b,i] Takuji Yamada[a]

[a]School of Life Science and Technology, Tokyo Institute of Technology, Tokyo, Japan
[b]Division of Cancer Genomics, National Cancer Center Research Institute, Tokyo, Japan
[c]Department of Computing, Faculty of Technology, University of Turku, Turku, Finland
[d]Endoscopy Division, National Cancer Center Hospital, Tokyo, Japan
[e]Department of Colorectal Surgery, National Cancer Center Hospital, Tokyo, Japan
[f]Institute for Advanced Biosciences, Keio University, Yamagata, Japan
[g]Intestinal Microbiota Project, Kanagawa Institute of Industrial Science and Technology, Kanagawa, Japan
[h]Transborder Medical Research Center, University of Tsukuba, Ibaraki, Japan
[i]Department of Cancer Genome Informatics, Graduate School of Medicine, Osaka University, Suita, Osaka, Japan

**ABSTRACT** Accumulating evidence indicates that the gut microbiome and metabolites are associated with colorectal cancer (CRC). However, the influence of surgery for CRC treatment on the gut microbiome and metabolites and how it relates to CRC risk in postoperative CRC patients remain partially understood. Here, we collected 170 fecal samples from 85 CRC patients pre- and approximately 1 year postsurgery and performed shotgun metagenomic sequencing and capillary electrophoresis-time of flight mass spectrometry-based metabolomics analyses to characterize alterations between pre- and postsurgery. We determined that the relative abundance of 114 species was altered postsurgery ($P < 0.005$). CRC-associated species, such as *Fusobacterium nucleatum*, were decreased postsurgery. On the other hand, *Clostridium scindens*, carcinogenesis-associated deoxycholate (DCA)-producing species, and its biotransformed genes (*bai* operon) were increased postsurgery. The concentration of 60 fecal metabolites was also altered postsurgery ($P < 0.005$). Two bile acids, cholate and DCA, were increased postsurgery. We developed methods to estimate postoperative CRC risk based on the gut microbiome and metabolomic compositions using a random forest machine-learning algorithm that classifies large adenoma or early-stage CRC and healthy controls from publicly available data sets. We applied methods to preoperative samples and then compared the estimated CRC risk between the two groups according to the presence of large adenoma or tumors 5 years postsurgery ($P < 0.05$). Overall, our results show that the gut microbiome and metabolites dynamically change from pre- to postsurgery. In postoperative CRC patients, potential CRC risk derived from gut microbiome and metabolites still remains, which indicates the importance of follow-up assessments.

**IMPORTANCE** The gut microbiome and metabolites are associated with CRC progression and carcinogenesis. Postoperative CRC patients are reported to be at an increased CRC risk; however, how gut microbiome and metabolites are related to CRC risk in postoperative patients remains only partially understood. In this study, we investigated the influence of surgical CRC treatment on the gut microbiome and metabolites. We found that the CRC-associated species *Fusobacterium nucleatum* was decreased postsurgery, whereas carcinogenesis-associated DCA and its producing species and genes were increased postsurgery. We developed methods to estimate postoperative CRC risk based on the gut microbiome and metabolomic compositions. We applied methods to compare the

Address correspondence to Takuji Yamada, takuji@bio.titech.ac.jp, or Shinichi Yachida, syachida@cgi.med.osaka-u.ac.jp.

The authors declare a conflict of interest. Dr. Shinji Fukuda and Dr. Takuji Yamada are founders of Metabologenomics. The company is focused on the design and control of the gut environment for human health. The company had no control over the interpretation, writing or publication of this work. The terms of these arrangements are being managed by Keio University and Tokyo Institute of Technology according to their conflict of interest policies.

estimated CRC risk between two groups according to the presence of large adenoma or tumors after 5 years postsurgery. To our knowledge, this study is the first report on differences between pre- and postsurgery using metagenomics and metabolomics data analysis. Our methods might be used for CRC risk assessment in postoperative patients.

**KEYWORDS** colorectal cancer, surgery, metagenomics, human gut microbiome, metabolomics

Colorectal cancer (CRC) is the third most common cancer, with over 1.8 million new cases and approximately 881,000 deaths per year worldwide (1). Its incidence has substantially increased over the past 5 years, accompanied by a gradual increase in CRC surgeries over the past 5 years in Japan (2). Postoperative patients are reported to be at an increased CRC risk (3–6). Therefore, quantitative evaluation of potential CRC risk in postoperative patients is strongly required.

Accumulating evidence indicates that the carcinogenesis and progression of CRC are linked to the gut microbiome (7, 8), in addition to genetic and other environmental factors (9). The gut microbiome and its metabolites not only reflect cancerous intestinal conditions (10–14) but also directly affect carcinogenesis (7, 15). For instance, our previous study showed that the abundance of several species and their estimated growth rates were elevated along with CRC progression (12). Several studies that compared CRC tissue with non-CRC mucosa from the same patients showed that most of the amino acids (e.g., serine) were elevated in CRC tissue (13, 14). Altogether, these results suggest that the presence of tumors, especially large malignant tumors in the advanced stage of CRC, and associated changes in cancerous intestinal conditions (e.g., stricture and inflammation) might lead to further enrichment of several bacteria and bacterial metabolites. In addition, several microbial activities might result in CRC carcinogenesis through host DNA damage. For example, deoxycholate (DCA), which is a secondary bile acid and can be biotransformed by a specific group of gut microbes (e.g., *Clostridium scindens* and *Clostridium hiranonis*), was also reported to be associated with liver cancer through DNA damage (15) and hypothesized to promote CRC carcinogenesis (16). Overall, revealing the relationship between the gut microbiome and/or metabolites and cancerous intestinal conditions is necessary for reducing the risk of microbe-derived CRC carcinogenesis (17).

In addition, the gut microbiota structure might be associated with postoperative outcomes (18). Despite the high risk of CRC in postoperative patients, endoscopic or surgical resection of tumors is still considered the only curative option for CRC treatment (19). Whereas there is an advanced understanding of the role of the gut microbiome in CRC carcinogenesis, the influence of surgery on the gut microbiome and metabolites remains partially understood. At this time, only a few studies have compared the gut environment pre- and postsurgery (20, 21). Ohigashi et al. reported alterations in several bacteria and organic acids (e.g., butyric acid) pre- and 7 days postsurgery using quantitative PCR (qPCR) and high-performance liquid chromatography (HPLC), respectively (20). Sze et al. revealed alterations in the gut microbiome using 16S rRNA sequencing between pre- and approximately 1 year postsurgery (21). The long-term effect of surgery on species-level gut microbiota, their genes, and other metabolites (e.g., bile acid) has yet to be elucidated.

Here, we collected 170 fecal samples from 85 CRC patients pre- and approximately 1 year postsurgery to investigate the long-term influence of surgery on the gut microbiome and metabolome. We characterized the fecal metagenomic and metabolomic alterations between pre- and postsurgery. Additionally, we developed methods for postoperative CRC risk assessment based on the gut microbiome and its genes and metabolomic compositions using a random forest algorithm that classifies large adenoma or early-stage CRC and healthy controls from our publicly available data sets (12). We then applied classifiers to preoperative samples and validated this method by comparing the estimated CRC risk between two groups according to clinical findings 5

**TABLE 1** Participants characteristics

| Characteristic | Data for patients who underwent: | |
| --- | --- | --- |
| | ESD[a] | Surgery |
| No. of participants with metagenome | 11 | 85 |
| No. of participants with metabolome | 9 | 83 |
| Days between colonoscopy (median $\pm$ SD) | 356 $\pm$ 69.4 | 374 $\pm$ 189.2 |
| CRC stage pretreatment (no. in stage 0, I, II, III, IV) | 11, 0, 0, 0, 0 | 2, 20, 37, 24, 2 |
| Colectomy (no. left/no.) | | 65/20 |
| No. of participants with chemotherapy | 0 | 22 |
| No. of patients with low postoperative CRC risk | 1 | 57 |
| No. of patients with high postoperative CRC risk | 3 | 19 |
| Tumor size pretreatment (median $\pm$ SD [mm]) | 24 $\pm$ 27.7 | 45 $\pm$ 18.9 |
| Serum cholesterol pretreatment (median $\pm$ SD [mg/dL]) | 210 $\pm$ 33.3 | 208 $\pm$ 40.5 |
| Serum cholesterol posttreatment (median $\pm$ SD [mg/dL]) | 195 $\pm$ 12.7 | 212 $\pm$ 38.1 |
| Age pretreatment (median $\pm$ SD [yrs]) | 69 $\pm$ 8.24 | 62 $\pm$ 9.63 |
| Gender (no. female/no. male) | 5/6 | 39/46 |
| BMI pretreatment (median $\pm$ SD [kg/m$^2$]) | 22.5 $\pm$ 3.15 | 22.5 $\pm$ 3.37 |
| Brinkman index pretreatment (median $\pm$ SD) | 390 $\pm$ 624 | 370 $\pm$ 410 |
| Alcohol intake pretreatment (median $\pm$ SD [g/day]) | 20 $\pm$ 421 | 0 $\pm$ 391 |

[a]ESD, endoscopic submucosal dissection; BMI, body mass index.

years postsurgery. Our findings and methods can be used for the CRC risk assessment in postoperative patients.

## RESULTS

**Patient characterization.** We collected 192 fecal samples from 96 CRC patients presurgery and approximately 1 year postsurgery ($n = 85$) or endoscopic submucosal dissection (ESD) treatment ($n = 11$); these groups included patients in our previous study's data set (12) (days 373 $\pm$ 182, median $\pm$ standard deviation [SD]) (see Materials and Methods, Fig. S1a in the supplemental material, Table 1, and Table S1). First, these 96 patients were classified into five groups (stage 0, stage I, stage II, stage III, and stage IV) according to the preoperative clinicopathologic findings. Eleven CRC patients (stage 0, $n = 11$) underwent ESD treatment. Eighty-five patients (stage 0, $n = 2$; stage I, $n = 20$; stage II, $n = 37$; stage III, $n = 24$; stage IV, $n = 2$) underwent surgical treatment (right-sided surgery, $n = 20$; left-sided surgery, $n = 65$). Furthermore, 22 CRC patients underwent chemotherapy after surgical treatment (stage III, $n = 21$, stage IV, $n = 1$). These 96 patients were then classified into the following three groups according to the presence of large polyps or tumors based on postoperative colonoscopy findings up to 5 years following surgery to characterize postoperative CRC risk in patients: (i) low postoperative CRC risk ($n = 58$), (ii) high postoperative CRC risk ($n = 22$), and (iii) unknown ($n = 16$) (see Materials and Methods; Fig. S1b; Table 1; Table S1).

**Surgical treatment alters the gut microbiome and metabolites.** We first verified whether the compositions of the gut microbiome and its genes and metabolites were changed between pre- and postsurgical treatment groups by permutational multivariate analysis of variance (PERMANOVA) based on the Bray-Curtis distance. The microbial genus profile was significantly different between the pre- and postsurgical treatment groups ($P = 1.00 \times 10^{-4}$) (Fig. 1a; Table S2). The Kyoto Encyclopedia of Genes and Genomes (KEGG) Orthology (KO) and metabolome profiles also showed similar patterns ($P = 1.00 \times 10^{-4}$ and $P = 2.00 \times 10^{-4}$, respectively) (Fig. 1b and c).

Next, we examined how much the gut microbiome and its genes and metabolites changed with surgical treatment by comparing Bray-Curtis dissimilarity between the same individuals. Because fecal samples from our healthy controls at different time points were not obtained, we downloaded publicly available metagenome (22) and metabolome (23) data from healthy control samples at different time points (see Materials and Methods and Table S1). The process for obtaining metagenome and

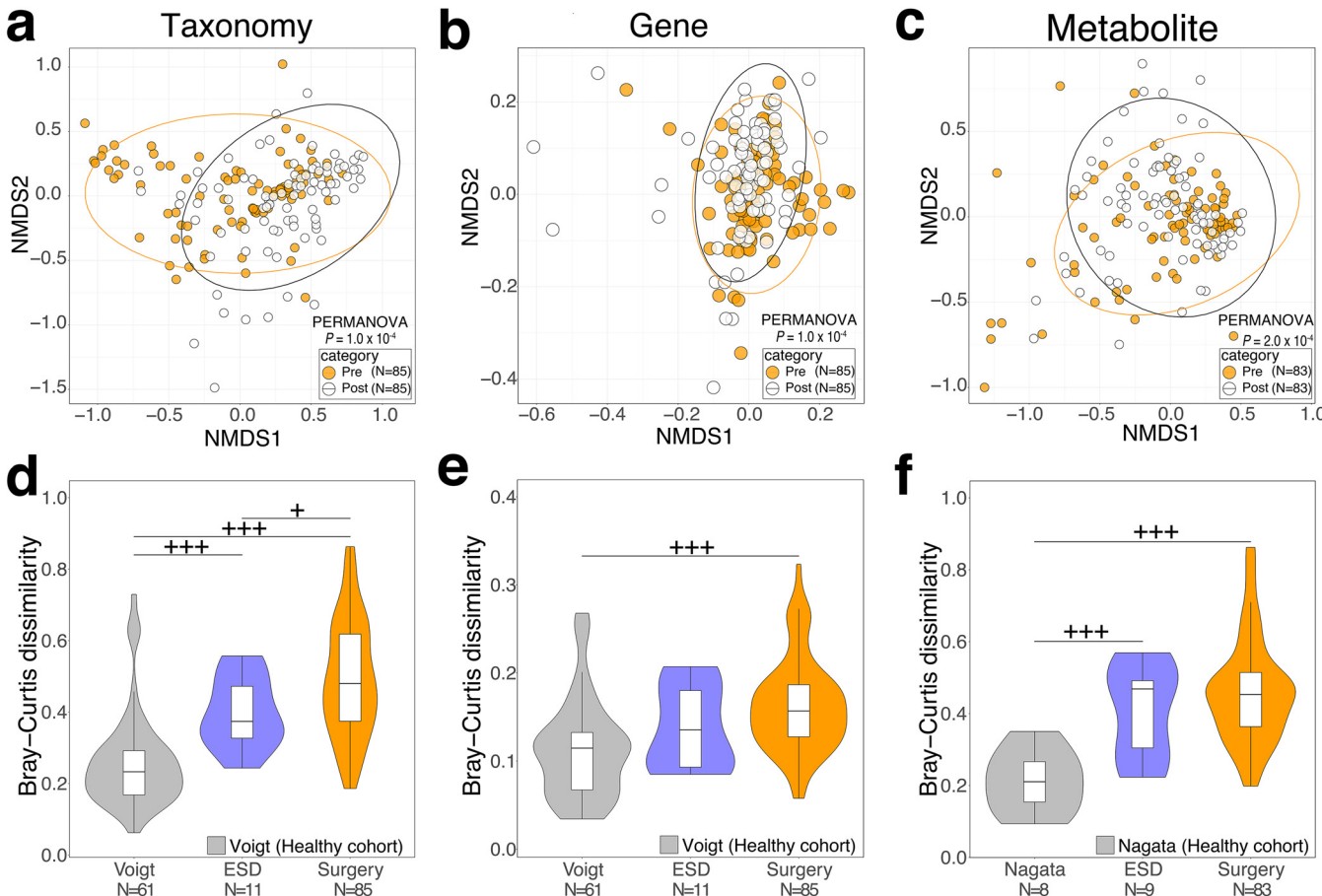

**FIG 1** Distinct fecal microbiome, gene, and metabolome compositions between pre- and postsurgical CRC patients. (a to c) NMDS analysis based on the Bray-Curtis distance was carried out to visualize the influence of surgical treatment on the overall community structure of the genus-level taxonomic profile (a), KO-level functional profile (b), and metabolome profile (c) in pre- and postsurgical treatment samples. *n* represents the number of samples. (d to f) Violin and box plots show the Bray-Curtis dissimilarity of the genus-level taxonomic profile (d), KO-level functional profile (e), and metabolome profile (f) of fecal samples at two different time points within the same participants. *n* represents the number of paired samples in the same individuals. The boxes in box plots represent 25th to 75th percentiles, black lines indicate the median, and whiskers extend to the lowest and highest values within 1.5 times the interquartile range. (d and e) The Voigt et al. data sets obtained from healthy individuals at two different time points were used as the control for taxonomic and functional profile analysis (22). (f) The Nagata et al. data sets were used as control data from a healthy individual cohort for metabolomic analysis (23). PERMANOVA shows a difference between presurgical and postsurgical treatment in community structure in each profile in panels a to c. Statistical analysis was performed by a one-sided Wilcoxon rank-sum test in panels d to f. Significant differences are denoted as follows: $+++$, elevation ($P < 0.005$); $++$, elevation ($P < 0.01$); $+$, elevation ($P < 0.05$); $---$, depletion ($P < 0.005$); $--$, depletion ($P < 0.01$); and $-$, depletion ($P < 0.05$).

metabolome data sets from the fecal samples (e.g., DNA extraction, sequencing platform, and quantification of metabolome) in these studies was almost the same as those in this study (Text S1). Thus, the effect of technical differences on comparing data sets from our study with those of these studies may be minimized. For taxonomic profiles, Bray-Curtis dissimilarity between the same patients pre- and postsurgery was significantly higher than that between healthy controls (one-sided Wilcoxon rank-sum test, $P = 5.44 \times 10^{-15}$) (Fig. 1d; Table S2). Genes and metabolome profiles also showed a similar pattern when we compared the surgical treatment group with healthy controls (one-sided Wilcoxon rank-sum test, $P = 1.21 \times 10^{-8}$ and $P = 1.45 \times 10^{-5}$, respectively) (Fig. 1e and f).

**Microbial alterations between pre- and postsurgical treatment.** We identified 114 species with relative abundances that were significantly ($P < 0.005$) different between pre- and postsurgical treatment, of which 90 species showed significantly decreased abundances, while 24 species showed significantly increased abundances (one-sided Wilcoxon signed-rank test) (Fig. 2a; Table S2).

We hypothesized that species that were elevated in each of the multistep CRC progression stages were decreased postsurgery. To assess this hypothesis, we compared healthy controls in our cohort (12) with patients with multistep CRC progression

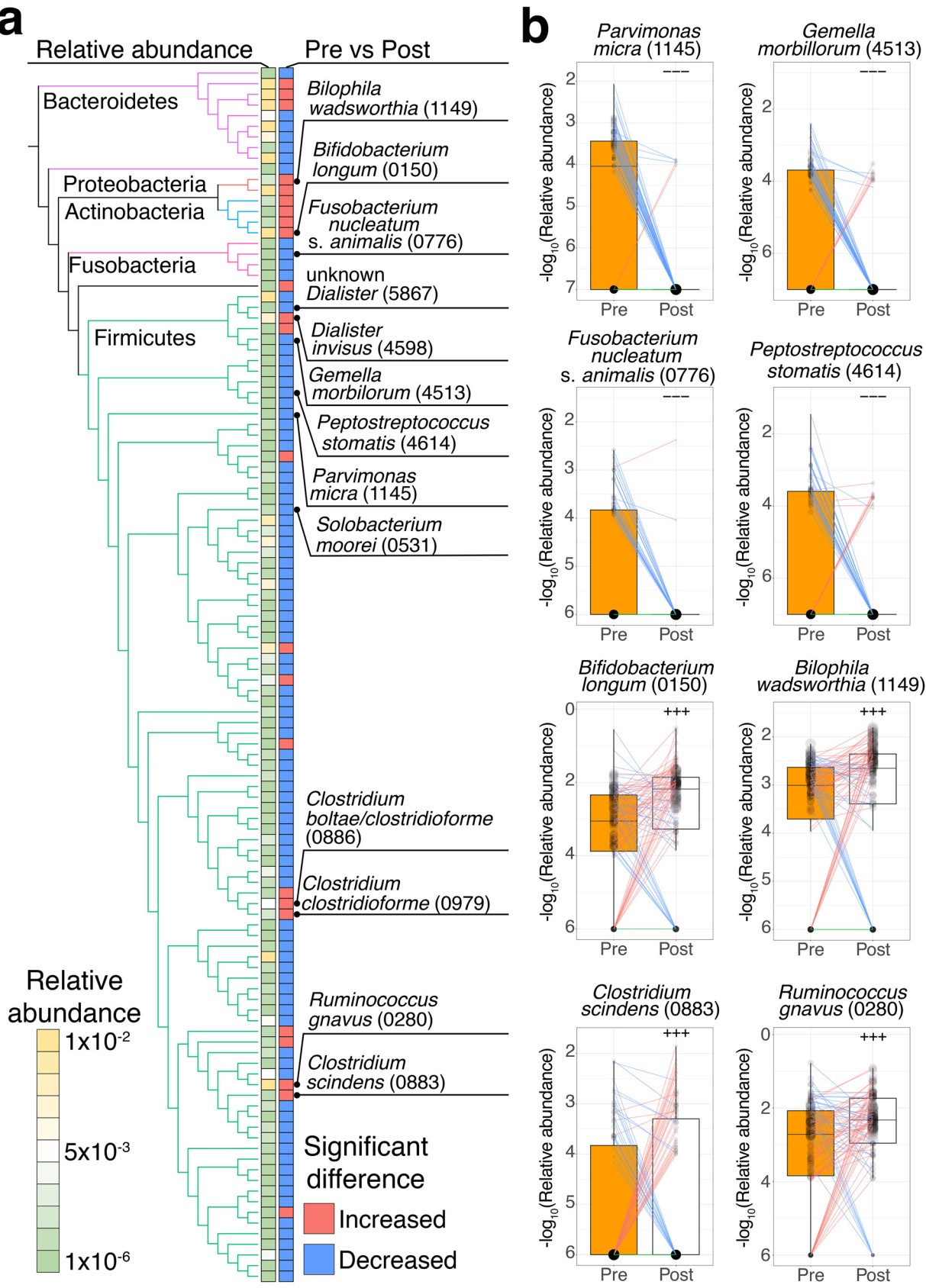

**FIG 2** Different enrichment patterns of the fecal microbiota between pre- and postsurgical treatment. (a) Comparison of the relative abundance of different species between the presurgical treatment ($n = 85$) and postsurgical treatment ($n = 85$) groups. The phylogenetic

(multiple polypoid adenomas with low-grade dysplasia [MP], stage 0, stage I/II, and stage III/IV) stages using a one-sided Wilcoxon rank-sum test. Ninety species were significantly ($P < 0.005$) elevated in at least one of the multistep CRC progression stages (one-sided Wilcoxon rank-sum test) (Table S2). Among the 90 significantly elevated species, 14 species were significantly decreased postsurgery. In particular, universal CRC markers in a meta-analysis of CRC (11), *Parvimonas micra* (ref_mOTU_v2_1145), *Gemella morbillorum* (ref_mOTU_v2_4513), *Fusobacterium nucleatum* subsp. *animalis* (ref_mOTU_v2_0776), and *Peptostreptococcus stomatis* (ref_mOTU_v2_4614) were significantly elevated in stage I/II and stage III/IV and decreased postsurgery (Fig. 2a and b). Unknown *Dialister* (meta_mOTU_v2_5867), unknown *Peptostreptococcaceae* (meta_mOTU_v2_5742), and *Solobacterium moorei* (ref_mOTU_v2_0531) species were also significantly elevated in stage I/II and stage III/IV and decreased postsurgery (Fig. 2a). These species were also reported as universal CRC markers in a meta-analysis of CRC (11).

Furthermore, we also examined 24 species that were significantly ($P < 0.005$) increased postsurgery (one-sided Wilcoxon signed-rank test) (Table S2). *Ruminococcus gnavus* (ref_mOTU_v2_0280), the isoDCA-producing species from DCA (24); *Bilophila wadsworthia* (ref_mOTU_v2_1149), a genotoxic hydrogen sulfide-producing species using taurine from taurocholate (25), and *C. scindens* (ref_mOTU_v2_0883), a genotoxic DCA-producing species (26), were also increased postsurgery (Fig. 2b).

**Changes in the estimated bacterial growth rate between pre- and postsurgical treatment.** We carried out an estimation of the growth rate in 24 species that showed significantly increased relative abundances postsurgery by using GRiD (27) and then compared it between pre- and postsurgery. Four species were elevated postsurgery (one-sided Wilcoxon rank-sum test, $P < 0.05$) (Table S2). Two species belonged to *Clostridium*, namely, *Clostridium clostridioforme* and *C. scindens*, and two species (*R. gnavus* and *Dialister invisus*) were elevated postsurgery (Fig. S2a).

Next, we investigated the association between bacterial relative abundance and estimated growth rate. The relative abundances of four species (*R. gnavus*, *C. scindens*, *Dialister invisus*, and *Bifidobacterium longum*) were positively correlated with their estimated growth rates in pre- and postsurgical treatment samples (both Spearman correlation coefficients $> 0.4$) (Table S2). On the other hand, the relative abundances of two species (*Flavonifractor plautii* and *B. wadsworthia*) were negatively correlated with their estimated growth rates in pre- and postsurgical treatment samples (both Spearman correlation coefficients $< -0.4$).

**Changes in the fecal metabolome profile between pre- and postsurgery.** A total of 60 metabolites were significantly ($P < 0.005$) altered between pre- and postsurgery (one-sided Wilcoxon signed-rank test) (Fig. 3a; Table S2). First, we focused on 43 metabolites that were significantly decreased postsurgery. Among them, 13 metabolites were significantly ($P < 0.005$) elevated in at least one of the multistep CRC progression stages, as previously reported (12) (one-sided Wilcoxon rank-sum test). In particular, one of the amino acids, serine, was significantly elevated in stages 0, I/II, and III/IV and decreased postsurgery (Fig. 3b). Furthermore, the amino acid-related metabolites Gly-Leu and urocanate were also significantly elevated in stages I/II and III/IV and decreased postsurgery (Fig. 3b). These metabolites were reported as advanced-stage CRC markers (12). The other amino acids, namely, methionine and cysteine, and the related metabolite, *N,N*-dimethylglycine, were also significantly decreased postsurgery (Fig. 3a).

**FIG 2** Legend (Continued)
tree and two heatmaps represent a phylogenetic relationship among 114 significantly different species, their average relative abundance in pre- and postsurgical treatment samples ($n = 170$, legend), and their significant differences between pre- and postsurgical treatment ($P < 0.005$; red, increase; blue, decrease), respectively. The edges in the phylogenetic tree represent five phyla (*Firmicutes*, green; *Fusobacteria*, pink; *Actinobacteria*, blue; *Proteobacteria*, orange; *Bacteroidetes*, purple). (b) Each box plot shows the $-\log_{10}$-transformed relative abundances of species in pre- ($n = 85$) and postsurgical treatment ($n = 85$) samples (pretreatment, orange; posttreatment, white). Each line in the box plot shows alteration patterns between pre- and postsurgical treatment within samples derived from the same patients (increase, red; decrease, blue; neither increase nor decrease, green). The sizes of points in the box plot reflect the distribution of the population in each category. Significant differences characteristics are denoted as follows: $+++$, increase ($P < 0.005$); $---$, decrease ($P < 0.005$). One-sided Wilcoxon signed-rank test was performed to characterize the increasing or decreasing trend in the pre- and postsurgical groups in panels a and b.

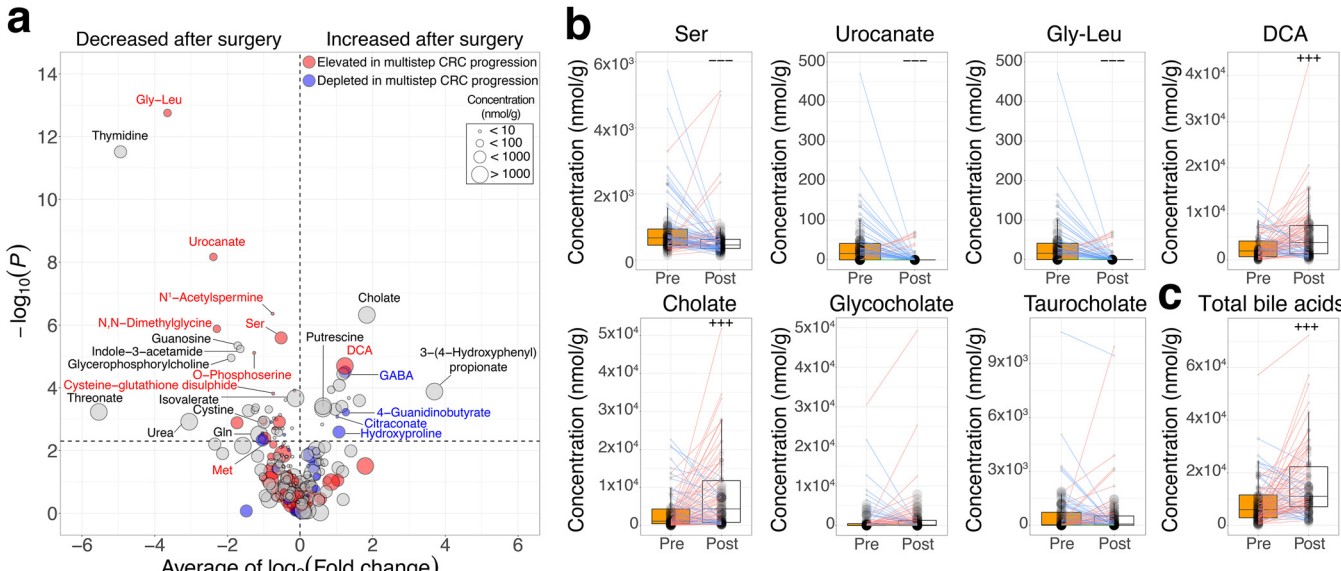

**FIG 3** Different enrichment patterns of fecal metabolites between pre- and postsurgical treatment. (a) Comparison of metabolite concentrations between the presurgical treatment ($n = 83$) and postsurgical treatment ($n = 83$) groups. The $x$ axis represents the average $\log_2$-transformed fold change between pre- and postsurgical treatment, while the $y$ axis represents the $-\log_{10}$-transformed $P$ values obtained from the one-sided Wilcoxon signed-rank test. The horizontal dashed line shows a $-\log_{10}$-transformed $P$ value of 0.005. The sizes of the circles in the legend represent the average concentration of each metabolite in pre- and postsurgical treatment samples. Elevation or depletion in at least one of the multistep CRC progression stages is colored red and blue, respectively (see Materials and Methods). (b and c) Each box plot shows the concentration of each metabolite (b) and total bile acids (c) in pre- ($n = 85$) and postsurgical treatment ($n = 85$) samples (pretreatment, orange; posttreatment, white). Each line in the box plot shows alteration patterns between pre- and postsurgical treatment within samples derived from the same patients (increase, red; decrease, blue; neither increase nor decrease, green). The sizes of points in the box plot reflect the distribution of the population in each category. A one-sided Wilcoxon signed-rank statistical test was performed to characterize the increasing or decreasing trend in the pre- and postsurgical groups. Significant differences characteristics are denoted as follows: $+++$, increase ($P < 0.005$); $---$, decrease ($P < 0.005$).

We also investigated 17 metabolites that were significantly ($P < 0.005$) increased postsurgery (one-sided Wilcoxon signed-rank test) (Table S2). Interestingly, DCA, which is well-known to be associated with CRC carcinogenesis (16), and cholate, which can be biotransformed to DCA, were significantly increased postsurgery (Fig. 3b).

Finally, we focused on the alteration of the other bile acid-related metabolites between pre- and postsurgical treatment. We examined changes in taurocholate and glycocholate between pre- and postsurgical treatment because these metabolites can be biotransformed to cholate by bacterial metabolism. The levels of taurocholate and glycocholate were not different between pre- and postsurgery (Fig. 3b) (Wilcoxon signed-rank test, $P = 0.568$ and $P = 0.202$, respectively). Next, we calculated the amount of total bile acids as the sum of taurocholate, glycocholate, cholate, and DCA and then compared it between pre- and postsurgery. The amount of total bile acids can be evaluated as the biosynthesis of bile acids or reabsorption capacity. The total bile acid content was significantly increased after surgery (one-sided Wilcoxon signed-rank test, $P = 1.86 \times 10^{-7}$) (Fig. 3c; Table S2). Moreover, we calculated the ratio of DCA to cholate and then compared it between pre- and postsurgery. This ratio may reflect the biotransformation rate to DCA from cholate. As a result, there was no significant difference between pre- and postsurgery (one-sided Wilcoxon signed-rank test, $P = 0.396$) (Fig. S3a; Table S2).

**Microbial gene alterations between pre- and postsurgery.** We focused on the genes associated with the biosynthesis of cholate and DCA because the expression of these genes was significantly increased postsurgery.

First, we investigated alterations of bile salt hydrolase (K01442), which can biotransform taurocholate/glycocholate to cholate. The relative abundance of bile salt hydrolase was not significantly different between pre- and postsurgery (Wilcoxon signed-rank test, $P = 0.277$) (Fig. 4a; Table S2).

Next, we examined alterations in the *bai* operon, which can biotransform cholate to

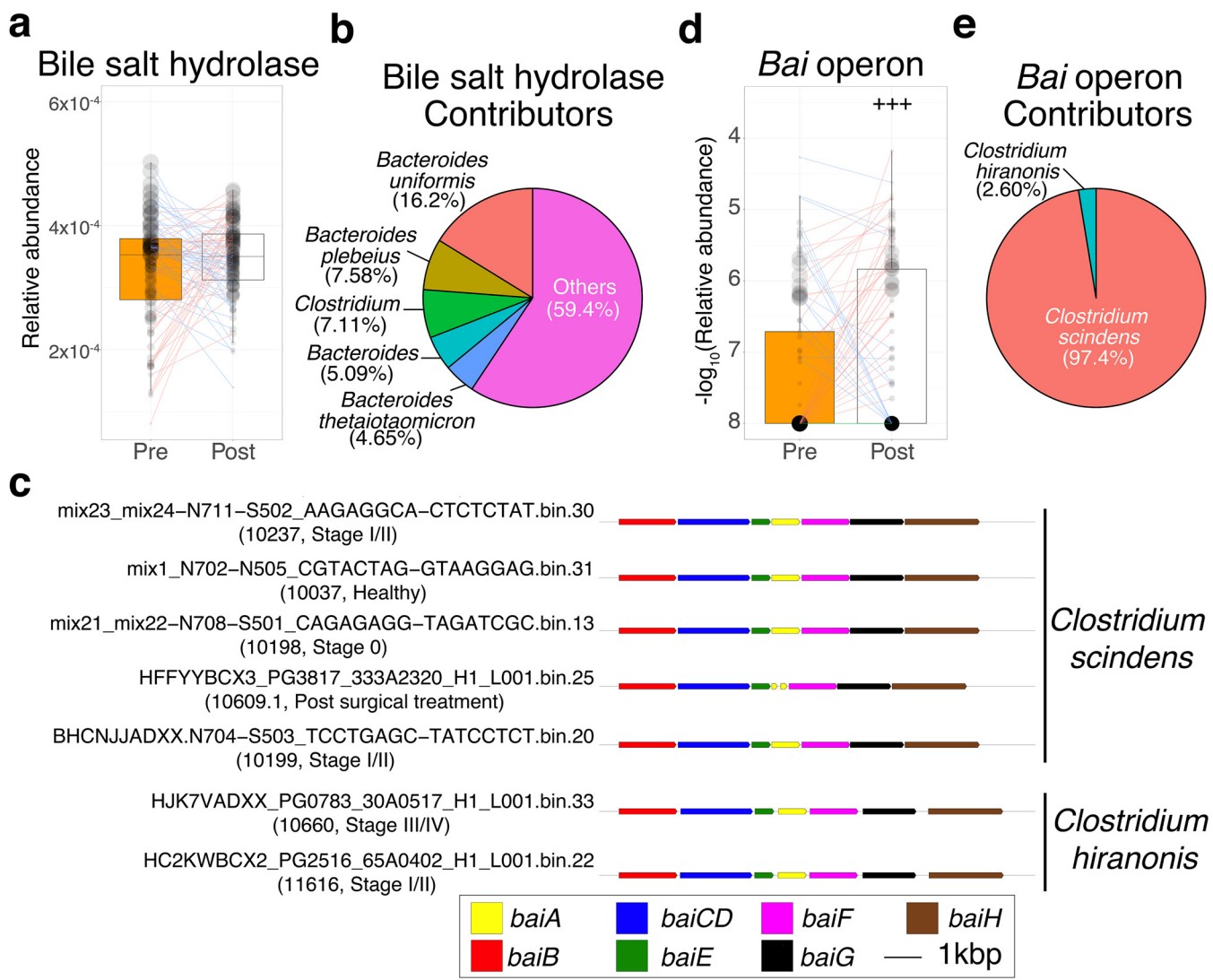

**FIG 4** Distinct enrichment patterns of bile acid-related genes between pre- and postsurgical treatment. (a and d) Each box plot shows the relative abundances (a) or $-\log_{10}$-transformed relative abundances (d) of genes or operons, respectively, in pre- ($n = 85$) and postsurgical ($n = 85$) treatment groups (pretreatment, orange; posttreatment, white). Each line in the box plot shows alteration patterns between pre- and postsurgical treatment groups within the samples derived from the same patient (increase, red; decrease, blue). The sizes of points in the box plot reflect the distribution of the population in each category. A one-sided Wilcoxon signed-rank statistical test was performed to characterize increasing or decreasing trends by comparing pre- and postsurgical treatment groups. Significant differences are denoted as follows: $+++$, increase ($P < 0.005$); $---$, decrease ($P < 0.005$). (b and e) Each pie chart shows the percentage of contributors to genes or operons. The percentages of contributors are indicated in parentheses. (c) Gene arrangement of the *bai* operon in seven metagenome-assembled genomes (MAGs). Samples are shown, and their category is indicated in parentheses. The colors in the legends reflect the gene name. Taxonomic assignments for MAGs were carried out by GTDB-Tk (see Materials and Methods).

DCA. A previous study reported that the *bai* operon is composed of eight genes, of which six genes (*bai B*, *bai CD*, *bai E*, *bai F*, *bai A*, and *bai H*) are required for the biotransformation of DCA to cholate (28). We applied a previously described similar approach (11), a metagenome annotation framework based on hidden Markov models (HMMs) (29) and homology search to identify and quantify the *bai* operon from the metagenome-assembled genomes (MAGs)-based gene catalog (see supplemental information). The MAG-based gene catalog has advantages for accurate annotation of genes based on gene arrangement and quantification of gene abundance based on operons. First, we identified 11 DCA-producing species that contain six *bai* genes in 22,547 MAGs derived from 716 metagenomic samples (cutoffs for quality of genomes, completeness > 50; contamination < 10) (Table S3). Among them, seven MAGs consisted of six *bai* genes in the same contig (Fig. 4c). Next, the quantification of each *bai* gene was carried out using the sum of each gene derived from those seven MAGs.

Finally, we compared the abundance of the *bai* operon between pre- and postsurgery. As a result, the *bai* operon was significantly increased postsurgery (one-sided Wilcoxon signed-rank test, $P = 4.88 \times 10^{-3}$) (Fig. 4d; Table S2). Additionally, a majority of the contributors were *C. scindens* (Fig. 4e).

**Influence of differences in the type of operation (right-sided versus left-sided) on bile acid metabolism.** Right-sided surgery involved resection of not only the right colon but also part of the terminal ileum (Fig. 5a and b), which might result in the inhibition of bile acid reabsorption (30). Thus, we also investigated the influence of the difference in operations (right-sided versus left-sided) on bile acid metabolism (right-sided, $n = 20$; left-sided, $n = 65$).

First, we examined the differences in the alterations in secondary bile acid (DCA), primary bile acids (cholate, glycocholate, and taurocholate), the ratio of DCA to cholate, and total bile acids between pre- and postsurgery for different operation types. DCA was increased in both patients who underwent right-sided surgery and patients who underwent left-sided surgery (one-sided Wilcoxon signed-rank test, $P = 0.0301$ and $P = 4.00 \times 10^{-5}$, respectively) (Fig. 5d; Table S2). Cholate was also significantly increased in patients after both right-sided and left-sided surgery (one-sided Wilcoxon signed-rank test, $P = 1.68 \times 10^{-4}$ and $P = 5.31 \times 10^{-4}$, respectively) (Fig. 5d). Glycocholate was significantly increased in patients after right-sided surgery (one-sided Wilcoxon signed-rank test, $P = 1.54 \times 10^{-3}$) (Fig. 5d). Taurocholate was increased in patients after right-sided surgery, whereas it tended to be decreased in patients after left-sided surgery (one-sided Wilcoxon signed-rank test, $P = 0.0327$ and $P = 0.0366$, respectively) (Fig. 5d). The concentration of total bile acids was significantly increased in patients after both right-sided and left-sided surgery (one-sided Wilcoxon signed-rank test, $P = 3.62 \times 10^{-7}$ and $P = 1.30 \times 10^{-4}$, respectively) (Fig. 5e).

Next, we investigated the differences in the alterations in bile acid-related genes and *C. scindens*, the major contributor of the *bai* operon, and *R. gnavus*, the isoDCA-producing species from DCA, between pre- and postsurgery for different operation types. Bile salt hydrolase was not significantly different between patients who underwent right-sided surgery and left-sided surgery (Wilcoxon signed-rank test, $P = 0.956$ and $P = 0.179$, respectively) (Fig. 5f; Table S2). The relative abundances of the *bai* operon and *C. scindens* were significantly increased in patients who underwent left-sided surgery (one-sided Wilcoxon signed-rank test, $P = 1.90 \times 10^{-3}$ and $P = 4.85 \times 10^{-3}$) (Fig. 5g and h). The relative abundance of *R. gnavus* was also increased in patients after both right-sided and left-sided surgery (one-sided Wilcoxon signed-rank test, $P = 0.0172$ and $P = 0.0344$) (Fig. 5i). The estimated growth rates of *C. scindens* and *R. gnavus* were higher in patients after left-sided surgery than before left-sided surgery (one-sided Wilcoxon rank-sum test, $P = 0.0360$ and $P = 0.0232$, respectively) (Fig. S2b).

**Effects of the chemotherapy as a potential confounder on the microbial and metabolome alterations between pre- and postsurgery.** Chemotherapy can alter host and microbial metabolism in the gut (31), which might be a potential confounder for investigation of the effect on surgical treatment. To determine whether the gut microbiome and metabolome alterations pre- and postsurgery are not associated with the chemotherapy, we investigated the gut microbiome and metabolome alterations pre- and postsurgery in 63 patients who underwent surgery without chemotherapy. As a result, 76 species and 35 metabolites were significantly ($P < 0.005$) altered between pre- and postsurgery (Table S2). Among them, alterations of 73 species and 35 metabolites showed the same patterns as those between pre- and postsurgery in 85 patients who underwent surgery or chemotherapy after surgery. These findings suggested that the majority of the gut microbiome and metabolome alterations between pre- and postsurgery are not associated with chemotherapy use.

**Estimation of postoperative CRC risk based on the gut microbiome and metabolome profiles.** In our previous study, we identified metagenomic and metabolomic signatures of CRC progression and examined their potential as CRC diagnostic biomarkers (12). These signatures can be used in a statistical framework to calculate

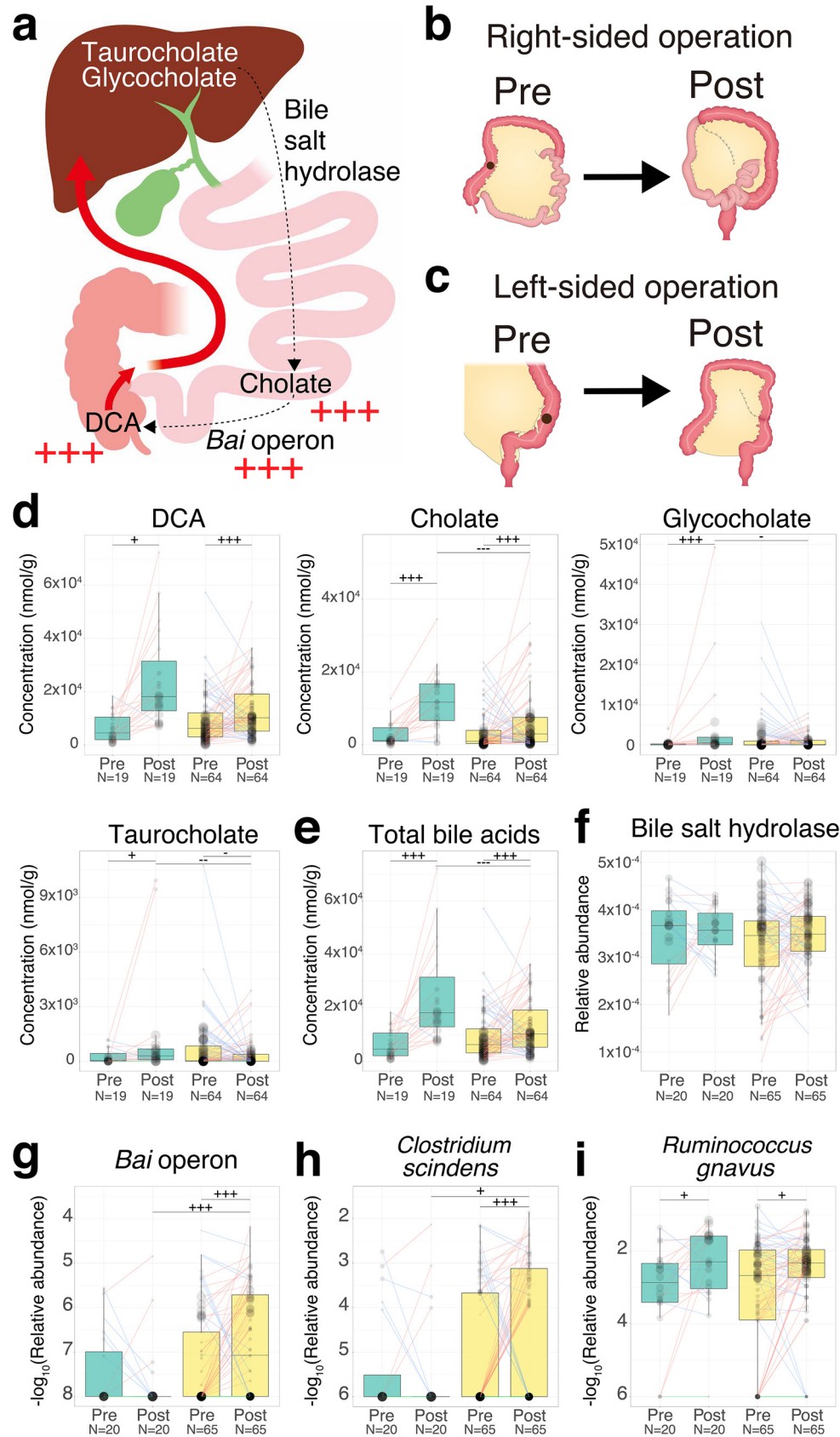

**FIG 5** Influence of right- or left-sided operations on bile acid metabolism. (a) Overview of bile acid metabolism in the gastrointestinal tract. Metabolites were placed corresponding to their production sites. The black arrow

the CRC probability of a new patient with CRC (10, 11, 32). We refer to CRC probability as "normalized probability" here.

We examined the potential of this statistical framework to estimate the postoperative CRC risk. First, we classified the CRC risk of postoperative patients as low or high risk according to the presence of large polyps or tumors based on postoperative colonoscopy findings for approximately 5 years following surgery (see Materials and Methods). Next, we built the stage-specific random forest-based binary classifiers to distinguish cases in each of the stages (MP, $n = 61$; stage 0, $n = 57$; stage I/II, $n = 85$; stage III/IV, $n = 59$) from healthy controls (H, $n = 245$) (see Materials and Methods and Table S4). We then applied these classifiers to 83 pre- and 83 postoperative metagenomic and metabolomic samples to obtain normalized probability in each sample. Finally, we investigated the potential of classifiers in estimating the postoperative CRC risk by comparing the normalized probability of the groups at low postoperative CRC risk ($n = 56$) with those of the groups at high postoperative CRC risk CRC ($n = 18$). In application of these classifiers to the preoperative samples, the normalized probabilities derived from MP and stage 0 classifiers were higher in the groups at high postoperative CRC risk than the groups at low postoperative CRC risk (one-sided Wilcoxon rank-sum test, $P = 0.0240$ and $P = 0.0129$, respectively) (Fig. 6a). In the application of these classifiers to the postoperative samples, the normalized probabilities derived from any classifiers were not different in the group at high postoperative CRC risk and the group at low postoperative CRC risk (one-sided Wilcoxon rank-sum test, $P > 0.05$) (Fig. 6b).

We investigated the contributors in each classifier that could estimate the signatures of postoperative CRC risk (see Materials and Methods). Several amino acids (e.g., Ala, Tyr), bile acids (DCA), and bile acid-related metabolites (e.g., taurine) and genes (ggt, K00681, glutamyl transpeptidase) were identified as top metabolomic and metagenomic signatures in MP or stage 0 classifiers (Fig. 6c).

## DISCUSSION

The gut microbiome and its metabolites have been associated with CRC carcinogenesis (7) and CRC progression (8). However, the influence of surgical treatment for CRC on the gut microbiome and metabolome and how it is related to CRC risk in postoperative CRC patients remain incompletely understood. Here, we showed alterations in the gut microbiome and metabolome due to surgical treatment for CRC. Our study indicated that 14 species (e.g., *F. nucleatum*) and 13 metabolites (e.g., serine), which were elevated at least in one of the multistep CRC progression stages, were significantly decreased postsurgery. Previous studies demonstrated that *F. nucleatum*, a protumorigenic bacterium (33), could potentiate CRC cells (8). In addition, serine is an energy resource for the production of pyruvate in CRC tissue, which plays a role in CRC

**FIG 5** Legend (Continued)
represents biotransformation by bacterial metabolism. The red arrow represents the flow of bile acid reabsorption. The red plus sign represents the increase in metabolites or operons in the postsurgical samples compared to the presurgical samples (one-sided Wilcoxon signed-rank test, $P < 0.005$). (b) The right-sided surgery was defined by resection of not only the right colon but also part of the terminal ileum. (c) The left-sided surgery was defined by resection of part of the left colon. The black point represents the position of CRC in panels b and c. (d to i) Each box plot shows the concentration of each metabolite (d), total bile acids (e), the relative abundance of genes (f), the $-\log_{10}$ transformed relative abundance of operon (g), and the $-\log_{10}$ transformed relative abundance of species (h and i). The sizes of the points in the box plots reflect the distribution of the population in each category. The colors of boxes in the box plot represent the right (green)- or left (yellow)-sided surgery groups, which were based on the CRC location presurgery. Each line in the box plot shows alteration patterns between pre- and postsurgical treatment groups within samples derived from the same patient (increase, red; decrease, blue; neither increase nor decrease, green). The number of samples is represented at the bottom of each category. A one-sided Wilcoxon rank-sum statistical test was performed to characterize the elevation or depletion pattern between samples from before right- and left-sided surgery or after right- and left-sided surgery. A one-sided Wilcoxon signed-rank statistical test was performed to characterize the increase or decrease trend between samples from before and after right- or left-sided surgery. Significant difference characteristics are denoted as follows: $+++$, elevation or increase ($P < 0.005$); $++$, elevation or increase ($P < 0.01$); $+$, elevation or increase ($P < 0.05$); $----$, depletion or decrease ($P < 0.005$); $--$, depletion or decrease ($P < 0.01$); and $-$, depletion or decrease ($P < 0.05$).

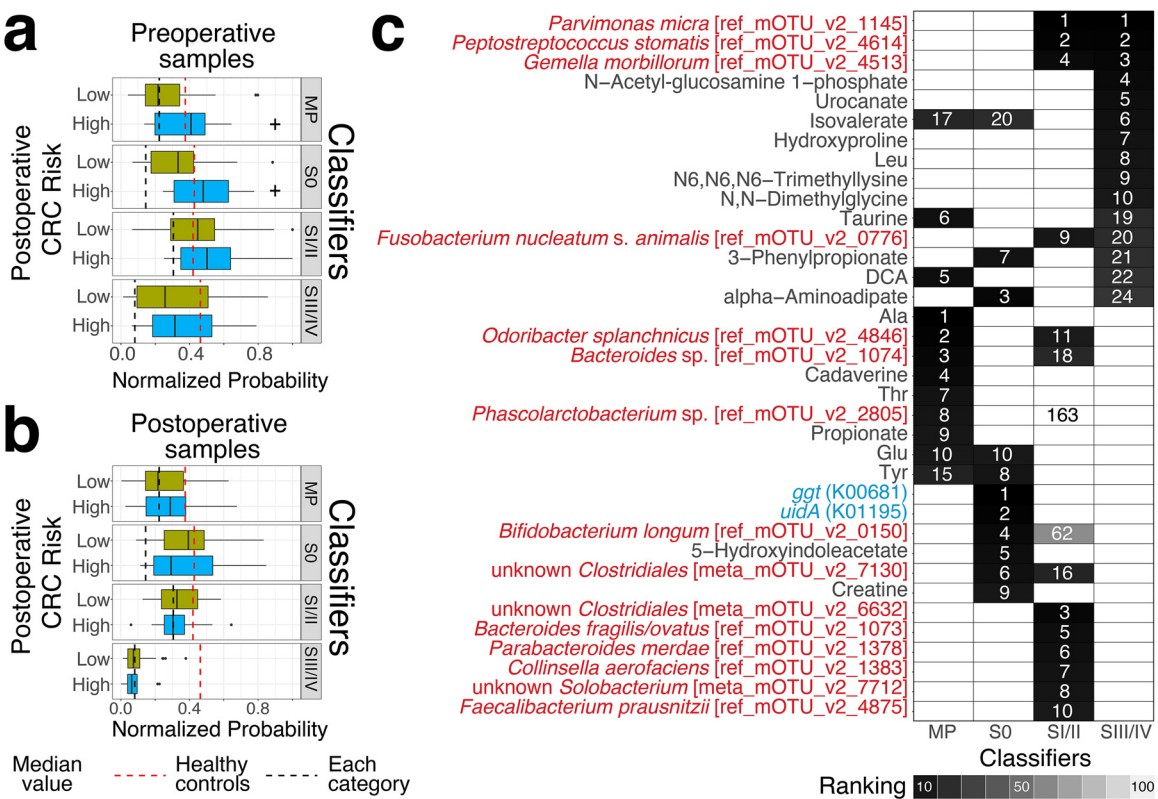

**FIG 6** Application of the classifier to samples obtained from pre- and postsurgical treatment. (a and b) We built four stage-specific random forest-based binary classifiers to distinguish cases in each of the stages (MP, $n = 61$; stage 0, $n = 57$; stage I/II, $n = 85$; stage III/IV, $n = 59$) from healthy controls (H, $n = 245$) (see Materials and Methods). We applied the classifiers to pre- and postoperative metagenomic and metabolomic samples from 76 patients with low and high postoperative CRC risk to obtain normalized probability (pre- and postoperative samples from patients at low postoperative CRC risk, $n = 58$; pre- and postoperative samples from high postoperative CRC risk patients, $n = 18$). The low and high CRC group classification was based on the postoperative colonoscopy findings for about 5 years following surgery. The box plots represent the normalized probability derived from each classifier in preoperative (a) and postoperative samples (b). Dashed lines show the median of normalized probability (red, healthy controls; each, MP). A one-sided Wilcoxon rank-sum statistical test was performed to characterize the elevation or depletion patterns in the group at postoperative CRC risk compared to the group at low postoperative CRC risk. Significant differences are denoted as follows: +, increase ($P < 0.05$). (c) Heatmap shows the rank of the top 10 contributors in each classifier (see Materials and Methods). The $y$ axis is ordered by the rank of contributors from the stage III/IV classifier. Color represents the category of contributors (species, red; KO, blue; metabolite, black). Color and the number in the heatmap represent the rank of contributors in each classifier (Legend).

cell growth (34). The species and metabolites that were decreased postsurgery, which were elevated in advanced-stage CRC, may reflect the improvement in cancerous intestinal conditions in postoperative patients. The decrease in other universal CRC markers derived from bacteria (11) (e.g., *P. micra*, *G. morbillorum*, and *P. stomatis*) and advanced-stage CRC markers derived from metabolites (12) (e.g., urocanate, Gly-Leu, and *N,N*-dimethylglycine) postsurgery supports the idea that surgery might improve cancerous intestinal conditions.

Tjalsma et al. proposed a bacterial driver-passenger model in CRC in which CRC carcinogenesis-associated species (i.e., drivers) may be transiently colonized in the initiating CRC development and replaced during CRC progression by other species (i.e., passengers) with a competitive growth advantage in the tumor niche environment (35). Few studies have been performed to formally test this hypothesis and distinguish CRC-associated species as passengers and/or drivers (36, 37). Despite some limitations, as discussed later, our results showed dramatically decreased CRC-associated species postsurgery, which might reflect an increase in the selective pressure for passenger species by removing the tumor during surgery. Together, our approach for comparing gut microbiota and metabolites pre- and postsurgery has the potential to identify the passengers that may also be drivers in CRC carcinogenesis.

In contrast, we also observed increased CRC carcinogenesis-associated metabolites, their producing genes, and their contributors in postsurgical patients. Our study

showed that DCA, its producing genes (*bai* operon), and its contributors (*C. scindens*) significantly increased postsurgery. Moreover, the estimated growth rate of *C. scindens* was also elevated postsurgery. Several studies have shown that DCA is associated with liver cancer (15) and hypothesized to promote CRC carcinogenesis (16). Altogether, DCA enrichment in postoperative patients might be associated with increasing CRC risk, suggesting that surgical treatment incompletely recovers cancerous intestinal conditions.

In addition, the alteration in bile acid metabolism between pre- and postsurgery may be affected by the difference in the type of operation. In particular, right-sided surgery involved not only resection of the right colon but also part of the terminal il-eum, which might result in the inhibition of bile acid reabsorption (30). The specific influence of the right-sided surgery is consistent with the increase in total bile acids, glycocholate, taurocholate, cholate, and DCA in patients after right-sided surgery. These findings were consistent with the high prevalence of diarrhea caused by the enrichment of bile acids in the colon in patients who underwent right-sided surgery (38). However, the concentration of total bile acids in fecal samples was also signifi-cantly increased in patients after left-sided surgery. There was no significant difference in total serum cholesterol, which reflects bile acid biosynthesis in the liver between before and after left-sided surgery (Wilcoxon signed-rank test, $P = 0.406$) (Table S2 in the supplemental material). The mechanism for the enrichment of total bile acids in left-sided surgical treatment is yet to be understood; however, we hypothesized that the decrease in bile acid reabsorption capacity might be caused by the surgical opera-tion in patients after left-sided surgery. In addition, we observed an increase in cholate and a decrease in taurocholate in patients after the left-sided surgery. Further work is needed to investigate the expression level of bile salt hydrolase to identify contribu-tors, as there was no difference in the amount of the associated gene between before and after left-sided surgery. The increase in *C. scindens* with the enrichment of cholate might result in an increase in DCA. The elevation of the estimated growth rate of *C. scindens* in patients after left-sided surgery and after right-sided surgery may support this idea. Trends for the decrease in the relative abundance of the *bai* operon and *C. scindens* in patients after right-sided surgery might be explained by the decrease in DCA-producing species in the right colon owing to resection (39). In fact, a positive correlation between the relative abundance of the *bai* operon or *C. scindens* and DCA concentration was observed only in patients after left-sided surgery (Spearman correla-tion coefficients after left-sided surgery, 0.354 and 0.350, respectively) (Fig. S4b and d). While it is not easy to interpret the gene/species abundance derived from metage-nome data as enzyme activity, these results may support the alteration of DCA with *C. scindens* alteration. Furthermore, the increase in DCA with the enrichment of *C. scin-dens* might result in an increase in the relative abundance and growth rate of *R. gna-vus*, the isoDCA-producing species from DCA. Collectively, these results suggested that modification of the cholate concentration in the colorectum could be a potential target to reduce DCA and its downstream concentration of bile acids.

Previous studies developed the statistical framework to classify healthy controls and CRC cases using a machine-learning algorithm based on the gut microbiome com-position (10, 11, 32). This framework could be used to diagnose CRC for new patients. Thus, we examined the potential of this statistical framework to estimate the postoper-ative CRC risk. Our results showed that the normalized probabilities, which were obtained by applying the MP and stage 0 classifiers to preoperative samples, were higher in the groups at high postoperative CRC risk than in the groups at low postop-erative CRC risk. These results suggest that MP or stage 0 classifiers can be used as a method to estimate the postoperative CRC risk. Moreover, normalized probabilities derived from these classifiers have a potential for reflecting the CRC carcinogenesis risk because these classifiers are based on signatures from MP and stage 0 (intramucosal carcinoma) patients, who are characterized by the absence of tumors. In fact, DCA, one of the carcinogenesis-associated metabolites, was identified as top metabolomic and

metagenomic signatures in the MP classifier. All together, these findings also support the idea that the normalized probability derived from MP or stage 0 classifiers can be used as a measurement to estimate the postoperative CRC risk. However, it is difficult to further characterize the clinical relevance of the risk to reflect metachronous CRC.

We acknowledge that our study also presents limitations. Our study could not reveal the effect of the combination of confounders such as chemotherapy and different operations (right-sided versus left-sided) on the gut microbiome and metabolome due to the small number of those patients. In addition, antibiotic usage may affect alterations in the gut microbiome and metabolome as hidden confounders. Patients who underwent surgery were treated with antibiotics within 2 days before and after surgical treatment to prevent infection by pathogenic bacteria. However, a previous study indicated that the majority of gut microbiome compositions recovered to preantibiotic baseline levels within 1.5 months after treatment (40). Considering the long-term length between sample collection in our study (days, 373 ± 182 [median ± SD]), antibiotic usage might not affect a majority of the observed differences. Furthermore, it is unknown how long the effect of the surgery may remain in postoperative patients. We investigated the fold change of 24 species and 17 metabolites pre- and postsurgery in 5 patients over a 2-year interval. The results indicated that 6 species and 9 metabolites were increased in patients for 2 years postsurgery (median of $\log_2$ fold change > 0) (Fig. S5). Further prospective studies in large populations of postoperative CRC patients are required to support these results.

In conclusion, our results show that the gut microbiome and metabolites dynamically change between pre- and postsurgery. Our findings showed that species and metabolites that were elevated in advanced-stage CRC were significantly decreased after surgery. However, DCA, which is known to be associated with CRC carcinogenesis, and its biosynthetic genes and producing species were significantly increased postsurgery. To uncover the detailed molecular mechanism underlying the effect of DCA in CRC carcinogenesis in postoperative patients, further follow-up studies using large populations of postoperative patients are required. Our estimation methods for postoperative CRC risk might be used for CRC risk assessment in postoperative patients.

## MATERIALS AND METHODS

**Study participants and fecal sample collection.** This study was conducted among participants undergoing total colonoscopy in the National Cancer Center Hospital, Tokyo, Japan, as mentioned in our previous study (12). Informed consent was obtained from the institutional review boards of each participating institute (National Cancer Center, 2013-244; Tokyo Institute of Technology, 2014018, Keio University, Shonan Fujisawa Campus, 78). We excluded patients with hereditary or suspected hereditary diseases (e.g., Lynch syndrome and familial adenomatous polyposis) from this study.

We collected 716 fecal samples from 620 participants, including our previous study (12) (Fig. S1a in the supplemental material). For our previous study, 576 fecal samples derived from 576 participants with no history of CRC surgery were classified into five groups according to colonoscopy and histological findings: (i) healthy (H, $n = 251$, no remarkable colonoscopy findings or few [up to two] small polyps [<5 mm]), (ii) multiple polyploid adenomas with low-grade dysplasia (MP, $n = 67$, more than three adenomas, mostly more than five adenomas), (iii) intramucosal carcinoma (stage 0 CRC, $n = 73$, polyploid adenoma[s] with high-grade dysplasia), (iv) stage I/II CRC ($n = 111$), and (v) stage III/IV CRC ($n = 74$) based on the Union for international Cancer Control (UICC) TNM Classification of Malignant Tumors. In addition, 28 fecal samples from 28 CRC patients approximately 1 year postsurgery were also collected. In total, 604 fecal samples from 576 participants were collected from our previous study. In this study, 112 fecal samples from 44 CRC patients presurgery and approximately 1 year postsurgery and 24 CRC patients approximately 1 year post-ESD/surgical treatment were newly collected to investigate the alteration of the gut microbiome and metabolome between pre- and postsurgical treatment in detail. Including our previous study and in this study, we collected 170 fecal samples from these 85 CRC patients pretreatment and approximately 1 year after surgical treatment. In addition, we also collected 22 fecal samples from 11 CRC patients pretreatment and approximately 1 year post-ESD treatment. Overall, 192 fecal samples were paired (pre- and post-ESD/surgical treatment) within 96 CRC patients (Table S1).

The 96 fecal samples derived from 96 postoperative patients following approximately 1 year postsurgery were also classified into three groups according to postoperative colonoscopy findings as follows: (i) postoperative healthy (PH, $n = 81$, no remarkable, colonoscopy findings or few [up to two] small polyps [< 5 mm]), (ii) postoperative multiple polyploid adenomas with low-grade dysplasia (PMP, $n = 14$, more than three adenomas, mostly more than five adenomas), and (iii) postoperative intramucosal carcinoma (PS0, postoperative stage 0, $n = 1$, polyploid adenoma[s] with high-grade dysplasia).

To characterize postoperative CRC risk in postoperative patients, we reviewed the colonoscopy findings of 96 postoperative patients over 4 years of follow-up from approximately 1 year postsurgery to approximately 5 years postsurgery. Among them, these patients were classified as having PH ($n = 67$) or PMP ($n = 11$) according to the colonoscopy findings except for 18 patients who could not be assessed. Finally, 96 patients were also classified into three groups according to the presence of polyps or tumors based on postoperative colonoscopy findings for approximately 5 years following surgery as follows: (i) low postoperative CRC risk ($n = 58$, PH for approximately 5 years following surgery), (ii) high postoperative CRC risk ($n = 22$, PMP or PS0 even once for approximately 5 years following surgery), and (iii) unknown ($n = 16$, no follow-up colonoscopy findings in approximately 4 years of follow-up) (Fig. S1b).

Among the surgical treatment groups, 20 CRC patients underwent right-sided surgery, while 65 CRC patients underwent left-sided surgery. Furthermore, 22 CRC patients underwent chemotherapy after surgical treatment (patients in stage III/IV). Capecitabine, oxaliplatin, and the drug combination folinic acid, fluorouracil, and oxaliplatin (FOLFOX) were used for chemotherapy. None of the patients underwent radiation therapy.

Fecal samples were collected immediately at first defecation after the oral administration of bowel-cleansing agents at the hospital on the day of the colonoscopy (41). We then placed the fecal samples directly on dry ice and stored them at $-80°C$ for metagenomic and metabolomic analyses. Colonoscopy was performed after bowel cleansing.

**Whole-genome shotgun sequencing and quantification of fecal metabolites.** Whole-genome shotgun sequencing was carried out on the extracted genomic DNA from fecal samples (Text S1). Capillary electrophoresis-time of flight mass spectrometry (CE-TOF MS) was performed to quantify the fecal metabolites as previously described (42–44) (Text S1). We quantified 397 metabolites in 694 samples.

**Taxonomic and functional profiling.** We obtained high-quality reads from raw reads (Text S1; Fig. S1c).

High-quality reads were used to obtain taxonomic and functional profiles. Taxonomic profiles were generated by mOTUs2 profiler version 2.0.0 (45) (Text S1). In brief, this profiler can quantify taxonomic abundance at the species and genus levels by mapping 10 universal single-copy marker genes. Functional profiles were generated by our in-house pipeline. Briefly, we generated MAGs and their driven gene catalogs to quantify gene and KO (46) profiles (Text S1).

**Estimation of microbial replication rates.** False positives in the comparison of bacterial relative abundance could occur because alterations in the abundance of one bacterium could lead to changes in the relative abundances of other bacteria (47). Therefore, we estimated the replication rates using GRiD (27) version 1.2 with a single parameter. The basic algorithm is based on the calculation of the ratio between coverage at the peak (*ori*) and trough (*ter*) on the reference genome as the replication rate because the bacterial genome is bidirectionally replicated from the origin of replication (*ori*) to the terminus region (*ter*). GRiD can estimate the growth rate using the draft genome by reordering multiple contigs using these coverage depths at low sequencing coverage (coverage $> 0.2$).

The estimation of growth rate using GRiD required the reference genome and coverage of each species. Estimation of replication rates was successfully computed for 19 out of 24 species that had significantly ($P < 0.005$) increased relative abundance after surgical treatment (one-sided Wilcoxon signed-rank test). These metagenomic operational taxonomic unit (mOTU)-annotated species corresponded to their MAGs based on NCBI taxonomy and were utilized as reference genomes (Table S2).

For the statistical test, samples with low coverage (coverage $< 0.2$) were omitted because their replication rate could not be estimated. Among 19 species, two species were omitted (*Lactobacillus casei* and *Pyramidobacter piscolens*) owing to the low detection in total samples ($n = 120$ and $n = 62$ in 716 samples, respectively).

**Building the classifier.** To estimate the postoperative CRC risk of each metagenomic and metabolomic sample, we built a classifier based on the random forest algorithm (48) utilizing species, KOs, and metabolome profiles. Because our previous study showed that the gut microbiome and metabolome profiles varied according to the stage of CRC progression (12), we built four classifiers (healthy controls versus MP, stage 0, stage I/II, stage III/IV) using the RandomForestClassifier function from the sklearn.ensemble module in the Python package Scikit-learn version 0.19.1. We did not use metagenomic and metabolomic samples from pre- and postsurgical treatment to build the classifiers.

We estimated the CRC probability using metagenome and metabolome profiles from each classifier. We first estimated the CRC probability in each sample, which was utilized for building each classifier. Leave-one-out cross-validation was carried out by the LeaveOneOut function from sklearn.model_selection in Scikit-learn. Then, the CRC probability of each sample was estimated by the predict_proba function for each classifier. We then estimated the value in each sample that was not utilized for building each classifier by the predict_proba function for each classifier. We also estimated the CRC probability for pre- and postsurgical treatment samples.

The CRC probability might be affected by the performance of the classifier and the ratio of the number of samples between healthy controls and each category in each classifier. Thus, we computed the difference between the CRC probability and minimum CRC probability. The value was later divided by the difference between the maximum and minimum probability in each classifier to obtain the normalized probability.

Classifier contributors of each classifier were determined based on the RandomForestClassifier feature importance function sklearn.ensemble module from Scikit-learn.

For evaluation of the classifier performance, each classifier was validated by 10 randomized 10-fold stratified cross-validations, which were carried out by the StratifiedKFold function from sklearn.model_selection in Scikit-learn. The accuracy of each classifier was examined using the average area under the

concentration-time curve (AUC) in each test. We optimized the filter of average relative abundance or concentration, n_estimators, and max_depth hyperparameter to maximize the AUC in each classifier for species, KOs, and metabolome profiles (Table S4). For the combination model, we also optimized the n_estimators, max_depth, and number of contributors derived from each optimized classifier (Table S4).

**Visualization of the taxonomic tree.** A taxonomic tree based on the mOTUs2 marker genes was downloaded from https://motu-tool.org/data/mOTUs.treefile. We filtered out low-abundance mOTUs (average of mOTUs in all metagenomic samples $< 10^{-6}$) and nonsignificantly different mOTUs based on the statistical test between pre- and postsurgery (e.g., $P > 0.005$) from the taxonomic tree by the drop.tip function in the R package ape. The taxonomic tree in the species was later visualized by iTOL version 5.5 (49).

**Statistical analyses.** Low-abundance microbial features (species and KOs) were discarded (cutoffs of average relative abundance in 716 samples, species $< 0.00001$, KO $< 0.0000001$). The metabolites that were detected in less than 5% of 694 samples were also discarded.

To investigate the overall effect of ESD and surgical treatments on the gut microbiome and metabolome profiles, Bray-Curtis dissimilarity analysis within the same individual was carried out by the vegdist function parameter with the bray method in the R package vegan. PERMANOVA was also performed to investigate the difference between pre- and postsurgical treatment groups by the adonis function parameter with 9,999 permutations and the Bray method in the R package vegan.

Nonmetric multidimensional scaling (NMDS) was carried out to visualize the effects of surgical treatment by the metaMDS function parameters with 20 trymax from the R package vegan.

To identify the microbial feature alterations due to surgical treatment, a one-sided Wilcoxon signed-rank test was performed by the wilcoxsign_test function parameter with the exact distribution from the R package coin (Table S2). To identify CRC-associated microbial features, we compared healthy controls with each of the multistep CRC progression stage (MP, stage 0, stage I/II, and stage III/IV) groups using a one-sided Wilcoxon rank-sum test by the Wilcox_test function from the R package coin (Table S2).

A $P$ value of $<0.005$ was considered statistically significant in all statistical tests. Furthermore, all $P$ values were corrected by the Benjamini-Hochberg method, which is an estimation of the false-discovery rate (FDR), to obtain the FDR-corrected $P$ value ($q$ value) (Table S2).

**Data availability.** Nucleotide sequences are available in the DDBJ Sequence Read Archive (DRA) as DRA006684, DRA008156, and DRA011152. The cohort from Voigt et al. is available in the European Nucleotide Archive (ENA) database as ERP009422. The 22,547 MAGs derived from 716 metagenomic samples are available at http://matsu.bio.titech.ac.jp/CRC_MAGs/CRC_MAGs.tar.gz.

## SUPPLEMENTAL MATERIAL

Supplemental material is available online only.
**TEXT S1**, DOCX file, 0.03 MB.
**FIG S1**, PDF file, 0.2 MB.
**FIG S2**, PDF file, 2.4 MB.
**FIG S3**, PDF file, 1 MB.
**FIG S4**, PDF file, 1.3 MB.
**FIG S5**, PDF file, 0.5 MB.
**TABLE S1**, XLSX file, 0.1 MB.
**TABLE S2**, XLSX file, 3.1 MB.
**TABLE S3**, XLSX file, 0.01 MB.
**TABLE S4**, XLSX file, 0.01 MB.

## ACKNOWLEDGMENTS

We are thankful to all the participants and their families who participated in this study. We are thankful to Z. Nakagawa for inspiring discussions and I. Take, M. Sezawa, M. Iwahara, and M. Komori for expert technical assistance.

This work was supported in part by grants from the National Cancer Center Research and Development Fund (2020-A-4); the Japan Agency for Medical Research and Development (AMED) (JP18ek0109187 to S.F., S.Y., and T.Y.; JP19 cm0106464 to S.Y. and T.Y.; JP20gm1010009 to S.F.; JP20ck0106546 to Y.S., S.Y., and T.Y.; JP20 cm0106477 to S.S., S.Y., and T.Y.; and 20jk0210009 to T.S. and S.Y.); JST-ERATO (JPMJER1902 to S.F.); JST-AIP Acceleration Research (JPMJCR19U3 to S.Y. and T.Y.); Japan Society for the Promotion of Science (JSPS) KAKENHI (142558 and 221S0002 to T.Y., 18H04805 to S.F., and 20H03662 to S.Y.); the Food Science Institute Foundation (to S.F.); the Program for the Advancement of Research in Core Projects under Keio University's Longevity Initiative (to S.F.); Integrated Frontier Research for Medical Science Division, Institute for Open and Transdisciplinary Research Initiatives, Osaka University (to S.Y.); Joint Research Project of the Institute of Medical Science, the University of Tokyo (T.S. and S.Y.); the Takeda Science

Foundation (to S.Y. and S.F.); the Suzuken Memorial Foundation (to S.Y.); Yasuda Memorial Medical Foundation (to S.Y.); and Yakult Bio-Science Foundation (to S.Y.).

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
