## [Reviewer comments · mSystems]

Surgical treatment for colorectal cancer partially restores gut microbiome and metabolome traits

Hirotsugu Shiroma, Satoshi Shiba, Pande Erawijantari, Hiroyuki Takamaru, Masayoshi Yamada, Taku Sakamoto, Yukihide Kanemitsu, Sayaka Mizutani, Tomoyoshi Soga, Yutaka Saito, Tatsuhiro Shibata, Shinji Fukuda, Shinichi Yachida, and Takuji Yamada

Corresponding Author(s): Takuji Yamada, Tokyo Institute of Technology

Review Timeline:

Submission Date:

January 11, 2022

Accepted:

February 18, 2022

Editor: Nicholas Chia

Reviewer(s): Disclosure of reviewer identity is with reference to reviewer comments included in decision letter(s). The following individuals involved in review of your submission have agreed to reveal their identity: Scott Peterson (Reviewer #2)

Transaction Report:

DOI: <https://doi.org/10.1128/msystems.00018-22>

February 18, 2022

Dr. Takuji Yamada
Tokyo Institute of Technology
Tokyo
Japan

Re: mSystems00018-22 (Surgical treatment for colorectal cancer partially restores gut microbiome and metabolome traits)

Dear Dr. Takuji Yamada:

Your manuscript has been accepted, and I am forwarding it to the ASM Journals Department for publication. For your reference, ASM Journals' address is given below. Before it can be scheduled for publication, your manuscript will be checked by the mSystems production staff to make sure that all elements meet the technical requirements for publication. They will contact you if anything needs to be revised before copyediting and production can begin. Otherwise, you will be notified when your proofs are ready to be viewed.

Publication Fees:

We recognize that the video files can become quite large, and so to avoid quality loss ASM suggests sending the video file via <https://www.wetransfer.com/>. When you have a final version of the video and the still ready to share, please send it to mSystems staff at mssystemsjournal@msubmit.net.

For mSystems research articles, if you would like to submit an image for consideration as the Featured Image for an issue, please contact mSystems staff at mssystemsjournal@msubmit.net.

Sincerely,

Nicholas Chia

Editor, mSystems

Journals Department
Figure S4: Accept

Figure S3: Accept

Table S2: Accept

Figure S5: Accept

Figure S2: Accept

Supplemental text: Accept

Table S1: Accept

Table S4: Accept

Table S3: Accept

Figure S1: Accept